# Genome-Wide Analysis of Trehalose-6-Phosphate Phosphatase Gene Family and Their Expression Profiles in Response to Abiotic Stress in Groundnut

**DOI:** 10.3390/plants13081056

**Published:** 2024-04-09

**Authors:** Yue Liu, Xin Wang, Lei Ouyang, Ruonan Yao, Zhihui Wang, Yanping Kang, Liying Yan, Yuning Chen, Dongxin Huai, Qianqian Wang, Huifang Jiang, Yong Lei, Boshou Liao

**Affiliations:** 1Key Laboratory of Biology and Genetic Improvement of Oil Crops, Ministry of Agriculture and Rural Affairs, Oil Crops Research Institute, Chinese Academy of Agricultural Sciences, Wuhan 430062, China; liuyue199212@163.com (Y.L.); wangxin06@caas.cn (X.W.); oyl9121522521@163.com (L.O.); y604987275@163.com (R.Y.); wangzhihui0229@126.com (Z.W.); kangyanping@caas.cn (Y.K.); yanliying2002@126.com (L.Y.); chenyuning@caas.cn (Y.C.); dxhuai@163.com (D.H.); wqqwaityou@163.com (Q.W.); peanutlab@oilcrops.cn (H.J.); 2State Key Laboratory of Biocatalysis and Enzyme Engineering, School of Life Sciences, Hubei University, Wuhan 430062, China

**Keywords:** groundnut, trehalose-6-phosphate phosphatase, abiotic stress, expression analysis

## Abstract

Trehalose-6-phosphate phosphatase (TPP) is a pivotal enzyme in trehalose biosynthesis which plays an essential role in plant development and in the abiotic stress response. However, little is currently known about *TPPs* in groundnut. In the present study, a total of 16 *AhTPP* genes were identified, and can be divided into three phylogenetic subgroups. AhTPP members within the same subgroups generally displayed similar exon–intron structures and conserved motifs. Gene collinearity analysis revealed that segmental duplication was the primary factor driving the expansion of the *AhTPP* family. An analysis of the upstream promoter region of *AhTPPs* revealed eight hormone- and four stress-related responsive *cis*-elements. Transcriptomic analysis indicated high expression levels of *AhTPP* genes in roots or flowers, while RT-qPCR analysis showed upregulation of the six tested genes under different abiotic stresses, suggesting that *AhTPPs* play roles in growth, development, and response to various abiotic stresses. Subcellular localization analysis showed that AhTPP1A and AhTPP5A were likely located in both the cytoplasm and the nucleus. To further confirm their functions, the genes *AhTPP1A* and *AhTPP5A* were individually integrated into yeast expression vectors. Subsequent experiments demonstrated that yeast cells overexpressing these genes displayed increased tolerance to osmotic and salt stress compared to the control group. This study will not only lay the foundation for further study of *AhTPP* gene functions, but will also provide valuable gene resources for improving abiotic stress tolerance in groundnut and other crops.

## 1. Introduction

Trehalose is a non-reducing disaccharide which is commonly found in bacteria, algae, fungi, invertebrates, and high plants [1,2]. Trehalose metabolism plays an essential role in normal plant growth and development [3]. Trehalose 6-phosphate (T6P), the precursor of trehalose, is a central signaling molecule that regulates sucrose level and links plant development to its metabolic status [4,5,6]. Previous studies have demonstrated that T6P can delay leaf senescence in young plants by inhibiting SnRK1, an energy sensor protein kinase [7]. Moreover, trehalose metabolism is involved in protecting cell structures and bioactive molecules under adverse conditions, such as low temperature, drought, and high salinity [3]. Since plants generally accumulate only trace amounts of trehalose [8], it is implied that this sugar might not act as an osmoprotectant in vivo during stressful conditions. On the other hand, trehalose can regulate reactive oxygen species (ROS) accumulation and improve photosynthetic efficiency under abiotic stress [9]. For example, exogenous trehalose treatment has been shown to enhance photosynthesis in melon seedlings under cold stress [10], and it has also been found to improve heat tolerance in maize seedlings [11,12].

The biosynthesis of trehalose is catalyzed by a conserved, two-step metabolic pathway in plants. Firstly, glucosyl moiety from UDP-glucose is transferred to glucose 6-phosphate (G6P) to generate T6P via trehalose-6-phosphate synthase (TPS). Afterward, T6P is dephosphorylated into trehalose through the action of trehalose-6-phosphate phosphatase (TPP) [13]. With the advancement of high-throughput DNA sequencing technology, researchers have successfully identified *TPS* and *TPP* gene family members in the genomes of all major plant taxa [4,14,15]. This significant finding suggests that trehalose metabolism, regulated by these genes, is widely distributed and prevalent throughout the plant kingdom [2,16]. To date, researchers have discovered eleven *TPS* genes in the genomes of both Arabidopsis and rice. Moreover, Arabidopsis has been found to possess ten *TPP* genes, while rice has been found to have thirteen [17,18]. Plant TPP proteins contain a conserved phosphatase domain (TPP domain). All the ten Arabidopsis TPP enzymes (AtTTPA to AtTPPJ) appear to possess similar catalytic activities, but differ in their expression pattern, suggesting they have tissue- and/or development-specific functions *in planta* [4].

Several plant *TPP* genes participate in the abiotic stress response. For instance, the majority of wheat *TPP* genes were up-regulated after cold, salt, and/or drought treatment [19]. Genetic manipulation of plant *TPP* genes was studied as an effective approach to improve abiotic stress tolerance [4]. Knocking out *AtTPPI* in Arabidopsis rendered the plants more sensitive to drought. However, overexpressing *AtTPPI* resulted in drought resistance and increased water use efficiency by reducing stomatal apertures and improving root architecture [20]. In addition, overexpression of *OsTPP1* and *OsTPP3* improved resistance to cold and/or salt stress in rice [21,22]. Moreover, introducing rice *OsTPP1* into maize ears using a floral promotor increased yields under both normal and drought conditions [23].

Cultivated groundnut (also called peanut, *Arachis hypogaea* L. AABB, 2n = 40) is an important oilseed and cash crop, which supplies a rich source of vegetable oils and proteins for humans, and is grown worldwide [24,25]. The annual global production of peanuts is approximately 46 million tons (http://www.fao.org/faostat/en/#home, accessed on 1 November 2023). However, peanut growth is susceptible to various abiotic stresses, including extreme temperatures, high salinity, and drought, which can negatively affect its development and reduce crop productivity [26]. Therefore, it is vital to discover novel stress tolerance genes and apply them to molecular breeding. Since *TPPs* have been proven to play vital roles in plant abiotic stress tolerance, growth, and yield, they are considered potential targets for crop’s genetic improvement. To date, there has been no report about the identification of *TPP* genes in peanut, and their role in this crop’s development and stress resilience remain ambiguous. In this study, we identified a total of sixteen *TPP* genes (*AhTPPs*) in the peanut genome [27]. Their phylogenetic relationships, gene structures, duplication events, and expression patterns were comprehensively investigated. In addition, the heterologous expression of two *AhTPPs* in yeast enhanced salt and osmotic tolerance. These results not only highlight the importance of *AhTPPs* involved in crop development and abiotic stress, but also provide a solid basis for future investigations into their role in peanut.

## 2. Results

### 2.1. Identification of AhTPP Gene Family Members in Peanut

To identify *TPP* gene family members in *A. hypogaea* (cv. Tifrunner), the 23 Arabidopsis and rice TPP protein sequences, as well as the HMM file of the TPP domain were utilized as search queries in the Peanut Protein Database. After filtering out redundant results and confirming the presence of a TPP domain in the candidates, a total of 16 putative *TPP* genes were obtained in the cultivated peanut genome (Table 1). Due to the high sequence similarity observed among gene family members in peanut, which is an allotetraploid with two subgenomes (A and B), a naming convention was adopted. If the TPPs from the two subgenomes shared amino acid sequence identities exceeding 95%, they were assigned the same number, with “A” and “B” used to differentiate them. Consequently, the 16 *TPP* genes were named *AhTPP1A*-*AhTPP8B* based on their chromosomal positions and sequence similarities (Table 1). The deduced protein lengths of AhTPPs varied from 279 (*AhTPP7A*) to 445 amino acids (*AhTPP1B*), with molecular weights between 31.69 and 49.87 kDa. The calculated isoelectric points of the *TPP* genes varied between 5.49 (*AhTPP7B*) and 9.48 (*AhTPP4B*), with an average value of 8.23. The grand average of hydropathy (GRAVY) was calculated to be between −0.472 (*AhTPP8B*) and −0.256 (*AhTPP1B*), suggesting that all AhTPPs were hydrophilic. Based on subcellular localization analysis, it was determined that AhTPP proteins were predominantly localized in the cytoplasm and apoplasm.

### 2.2. Phylogenetic Analysis and Chromosomal Locations of the AhTPP Gene Family

The evolutionary tree analysis revealed that the 16 AhTPPs can be classified into three subgroups (I–III) (Figure 1A). Subgroups I and III comprised TPPs from all seven plant species, while subgroup II contained no TPP from Arabidopsis. There were eight, four, and four AhTPP members in subgroups I, II, and III, respectively. All peanut AhTPPs showed high homologies with AdTPPs and AiTPPs, which was consistent with the fact that *A. hypogaea* presumably originates from a hybridization of two diploid ancestors. In addition, most TPPs from the *Arachis* species exhibited close evolutionary relationships with those from *G. max*, implying that TPPs are conserved in legume plants.

According to the genome annotation information of *A. hypogaea*, we generated chromosomal distribution maps for the *AhTPP* gene family (Figure 1B). The results showed that the 16 *AhTPPs* are unevenly distributed on the chromosomes. For instance, there are three *AhTPP* genes on chromosome A08 (*AhTPP5A*, *AhTPP6A*, and *AhTPP7A*) and B07 (*AhTPP4B*, *AhTPP5B* and *AhTPP6B*), only one *AhTPP* is found on chromosome A05, A07, A10, B02, B05, and B10, while no *AhTPP* is located on the remaining ten chromosomes.

### 2.3. Gene Structure and Conserved Motif Analysis of the AhTPP Family

Gene structure analysis provided valuable information on the evolution of peanut *AhTPPs*. The number of exons *AhTPPs* varied from 7 to 14. Among them, *AhTPP5B* (group II) had the longest genomic length, measuring 12,741 bp. This can be attributed to the presence of a large-sized intron within the gene body (Figure 2B). In general, *AhTPPs* with close phylogenetic relationships tended to share similar exon–intron structures. For instance, members of subgroup I displayed similar exon lengths and distribution patterns. Conserved motif analysis of these sequences revealed that motif 1 was found in all identified AhTPPs, while motifs 2, 3, 4, 5, and 6 were distributed in most members. However, motif 9 was specifically identified in AhTPP5A and AhTPP5B (Figure 2C). Collectively, AhTPPs within the same subgroup exhibited great similarities in terms of gene structure and protein motifs (Figure 2A–C).

### 2.4. Collinear Analysis of AhTPP Genes

Gene duplication was a major driving force in the evolution and expansion of gene families [28]. Twelve paralogous pairs of *AhTPPs* were identified to have segmental duplication events, while no tandem duplication event was found in any *AhTPP* member (Figure 3A, Appendix A). These results suggested that segmental duplication was the main mechanism for the family’s expansion. The synonymous (Ks) and non-synonymous substitutions (Ka) rates of the duplicated *AhTPPs* were calculated using TBtools software (v1.129). The divergence time of duplicated gene pairs was estimated using the formula: T = Ks/2λ (λ = 8.12 × 10^−9^). The results indicated that segmental duplication probably occurred between 1.14 and 142.13 million years ago (Mya) (Appendix A). In addition, the ratio of Ka to Ks is an indicator of selective pressure for duplicated gene pairs, with Ka/Ks < 1 indicating purifying/negative selection and Ka/Ks > 1 Darwinian/positive selection. In this study, all Ka/Ks of the duplicated *AhTPP* gene pairs were < 1, indicating they might have undergone purifying selective pressure during evolution, and probably leading to limited functional divergence of the duplicated genes. A total of eight and nine *TPP* genes were identified in the two diploid ancestors, respectively (Appendix A). The collinearity analysis of *TPP* family members from three *Arachis* species showed that *A. hypogaea* had 11 orthologous *TPP* pairs with both *A. duranensis* and *A. ipaensis* (Figure 3B and Appendix A). To investigate the relationships of the TPP gene family among other model plants, we created syntenic maps comparing *AhTPPs* with those in *A. thaliana* and *G. hirsutum*. As shown in Figure 3C, both *A. thaliana* and *G. hirsutum* had 14 orthologous *TPP* pairs with *A. hypogaea* (Figure 3C and Appendix A).

### 2.5. Prediction of Cis-Acting Elements in the Promoter Region of AhTPPs

To further study the potential regulatory mechanism of *AhTPP* genes, sequences 1500 bp upstream of the *AhTPP* gene transcription start site were used to analyze *cis*-elements. A total of 15 different types of *cis*-elements were detected. Based on their functions, they were categorized into four main groups: plant hormone-responsive elements, stress-responsive elements, growth and development-related elements, and light-responsive elements (Figure 4; Appendix A). Eight types of plant hormone responsive elements were found in the promoters of *AhTPPs*, with ABRE (ABA responsiveness) being the most abundant. Four types of stress-related *cis*-elements were also identified, including TC-rich repeats (defense and stress responsiveness), LTR (low-temperature responsiveness), MBS (drought inducibility), and MBSI (regulation of flavonoid metabolism). These results implied that *AhTPPs* are possibly involved in multiple hormone and abiotic stress responses. A CAT-box element was found in the growth and development category. In the light responsiveness category, two *cis*-elements, namely G-box and MRE, were identified. G-box was found in 62.5% of *AhTPPs*, indicating that light signals may play a crucial role in the transcriptional regulation of *AhTPPs*.

### 2.6. Expression Profiles of AhTPP Genes in Different Tissues and under Abiotic Stress Treatments

Based on the analysis of transcript levels of *AhTPPs* in different developmental stages and tissues, it was found that all *AhTPP* genes, except *AhTPP7A*, were expressed in at least one tested sample (Figure 5A and Appendix A). Almost all *AhTPP* genes showed high expression in the perianth. Although the expression levels of *AhTPPs* were significantly in different tissues, *AhTPPs* within the same phylogenetic subgroup generally exhibited similar expression patterns. For instance, almost all *AhTPPs* in subgroup I showed substantially high expression levels in the roots and flowers, while members of subgroup III were preferentially expressed in the flowers (perianth and pistils). Specifically, seven paralogous pairs of *AhTPPs*, including *AhTPP1A*/*1B*, *AhTPP2A*/*2B*, had the same tissue-specific expression profiles.

RNA-seq data were obtained from the Peanut Genome Resource database to investigate the transcriptional response of *AhTPPs* to abiotic stress [29]. The expression levels of 10 *AhTPPs* were significantly upregulated in response to either drought or cold stress (Figure 5B and Appendix A). Moreover, according to our previous transcriptome data [30], nearly all *AhTPPs* showed increased expression levels at least at one time point (3, 24, 48 h) following cold treatment in SLH and/or ZH12 peanut cultivars (Figure 5C and Appendix A).

Since most plant *TPP* genes are induced by abiotic stress, based on the transcriptome results in Figure 5B,C, we selected six *AhTPP* genes, which were significantly up-regulated either by low-temperature or drought stress conditions, for further validation by RT-qPCR analysis. The results revealed that all the six *AhTPP* genes showed high expression in flowers, and among them, *AhTPP2A/2B* and *AhTPP6A* also exhibited relatively high expression in roots (Figure 6A), which is consistent with the transcriptional expression levels reported in the transcriptome databases. Subsequently, the transcriptional patterns of these six *AhTPPs* were validated under abiotic stress conditions. Almost all the analyzed *AhTPP* genes were upregulated in response to cold, salt, and/or drought stress at least at one tested time point. Although some *AhTPPs* showed very similar transcription profiles in both leaves and roots under stressful conditions, most genes had distinct expression levels between different tissues. For instance, the transcript level of *AhTPP1A* was enhanced and exhibited similar expression trends in leaves and roots in response to multiple abiotic stresses. In contrast, *AhTPP5A* had significantly different expression patterns in leaves and roots under the same stressful conditions (Figure 6B,C). Moreover, there were greatly increased levels (up to 500-fold) of *AhTPP5A* and *AhTPP8B* in the roots after cold, drought, and salt treatment, suggesting these genes were extremely susceptible to various environmental stresses (Figure 6C). Collectively, the RT-qPCR results are in agreement with previous transcriptome data, indicating that some *AhTPPs* could be used as potential targets to enhance abiotic stress resilience in peanut.

### 2.7. Subcellular Localization and Functional Characterization of AhTPPs

According to the expression data from RNA-seq and RT-qPCR results, we found that *AhTPP1A* was constitutively expressed in all 22 tested tissue samples and responded to cold and salt as well as drought stresses (Figure 6 and Appendix A). Moreover, the transcript level of *AhTPP5A* was sharply up-regulated by various abiotic stress. Given that the two *AhTPPs* (*AhTPP1A*, *AhTPP5A*) possibly play vital roles in peanut development and in the stress response, we selected them as candidates for further study. The subcellular localization results revealed that the green fluorescence signals of AhTPP1A and AhTPP5A fusion protein were observed in both the cytoplasm and nucleus. These signals partially overlapped with the red fluorescence signal of the nucleus marker. In contrast, the control GFP protein was evenly distributed throughout the entire cell. (Figure 7A). These findings suggest that the two AhTPPs are likely located in both cytoplasm and nucleus.

In our previous study, we successfully developed a highly efficient functional yeast screening system, specifically designed to identify potential abiotic stress tolerance genes in peanut [31]. Thus, the functions of the two *AhTPP* candidate genes were characterized via the yeast platform (Figure 7B). Transgenic yeast cells overexpressing *AhTPP1A* or *AhTPP5A* were able to grow on 1.0 M NaCl solid medium plates when they were at a 10^−2^ dilution level, while the negative control yeast containing the empty control (pYES2) could not survive under the same conditions. In addition, the transgenic yeast showed slightly more tolerance to mannitol-mediated osmotic stress than the empty vector with the concentration of mannitol increasing to 1.0 M, as no yeast cell with pYES2 was observed on the plate (1.0 M mannitol) when the cultures were diluted up to 10^−3^. These results demonstrate that the overexpression of either *AhTPP1A* or *AhTPP5A* leads to improved osmotic and salt tolerance in transgenic yeast.

## 3. Discussion

Trehalose metabolism plays a crucial role in plant growth, development, and abiotic stress responses. As a key regulator involved in trehalose synthesis, plant *TPPs* often occur in large gene families [4]. The rapid development of whole genome sequencing and the wealth of publicly available genomic data have allowed researchers to comprehensively identify the families of *TPP* genes in both monocots and dicots, including 10 in Arabidopsis, 11 in maize, 18 in soybean, and 30 in wheat. A previous study showed that *A. duranensis* and *A. ipaensis*, respectively, provided the A and B subgenomes of cultivated peanut [32]. In this study, we identified 16 *AhTPP* genes in cultivated peanut (Table 1), which appears to maintain copies of all *TPP* genes in its two diploid parents, suggesting that the *AhTPP* gene family has primarily expanded via genome-wide replication events. Besides, gene duplication events facilitate the generation of new genes and novel functions. During our analysis, we discovered 12 pairs of segmental duplicated *AhTPP* genes, while no tandem duplication event was observed (Figure 3A and Appendix A), indicating that segmental duplication was another major driver for the expansion of the *AhTPPs* family. Similar results were also found in soybean [33] and cotton [34]. Furthermore, the Ka/Ks ratio was less than one for all pairs of duplicated genes, indicating that *AhTPPs* underwent a stringent purifying selection (Appendix A). Therefore, these findings indicate that both whole-genome duplication and segmental duplication may play a significant role in the expansion and evolution of the *AhTPP* gene family.

According to the phylogenetic tree, the 16 identified AhTPPs were clustered into three distinct subgroups. There were four AhTPPs in subgroup II, while no Arabidopsis TPP member was found in this subgroup, suggesting the TPPs might undergo distinct selection pressures between peanut and Arabidopsis during evolution. Peanut has a specific growth habit of geocarpy, which means that the plant blooms above ground, while the fruit develops into a pod underground. Gene structure analysis indicated that the exon number of *AhTPP* genes ranged from 7 to 14, while the majority of the other plant *TPPs* possessed 9 to 11 exons. The large variation of exon numbers of *AhTPPs* probably generated diverse transcript variants that arise from alternative splicing events, which would facilitate peanut pod adaption to various environmental stresses in soil.

Generally, genes with similar structures and domains are likely to have similar functions. In this study, we observed that AhTPPs within the same phylogenetic subgroup had highly similar exon–intron arrangements and protein motifs (Figure 2), suggesting they might have conserved functions. Additionally, orthologous genes with high collinearity between different species usually have the same evolutionary origins and/or similar functions. In this study, both *A. thaliana* and *G. hirsutum* had 14 orthologous TPP pairs with *A. hypogaea* (Figure 3C and Appendix A), implying that they were likely to possess similar functions. For instance, a previous study showed that overexpression of *AtTPPD* enhanced plant tolerance to high salinity stress, as well as to increased starch levels and soluble sugar contents [35]. The collinearity between *AhTPP6B* and *AtTPPD* suggests that *AhTPP6B* might be involved in salt stress response, and could be used to improve salt resistance capability in peanut and other crops.

Since T6P is the direct substrate for TPPs in higher plants, TPP proteins are presumably located in the same subcellular compartments as T6P. In Arabidopsis, it was estimated that the majority of T6P (approximately 71%) was likely located in the cytosol and nucleus, and the remainder in the chloroplast and/or vacuoles [36]. In our study, bioinformatic prediction conducted using online tools suggested that most AhTPPs were located in the cytosol (Table 1), which overlaps with the localization of T6P. Additionally, the transient expression of the AhTPP-GFP fusion protein in tobacco protoplasts confirmed that AhTPP1A and AhTPP5A were likely localized both in the cytosol and nucleus. Similar results have also been observed in Arabidopsis, in that five AtTPP proteins (AtTPPA, AtTPPB, AtTPPC, AtTPPF, and AtTPPH) were targeted to both the cytosol and the nucleus [35]. This suggests that AhTPPs may play a significant role in peanut metabolism.

Although extensive studies have reported that plant *TPP* genes were expressed in various tissues, some of them showed tissue- and development-specific expression patterns. In our study, all the *AhTPPs* were observed to be expressed in the roots, but to different extents. For instance, *AhTPP3A/3B*, *AhTPP8A,* and *AhTPP7B* genes were expressed at very low levels, whereas others, such as *AhTPP1A/1B*, *AhTPP4A/4B*, and *AhTPP6A* were more highly expressed. Additionally, four AhTPPs in subgroup III (*AhTPP1A/1B*, *AhTPP3A/3B*) were preferentially expressed in the flowers (Figure 5A). Similar spatiotemporal *AtTPPs* expression patterns have also been found in Arabidopsis. For example, most *AtTPPs* were detected in the roots, but the expression levels of *AtTPPA*, *AtTPPF,* and *AtTPPG* (all belonged to subgroup III) were predominantly higher in the flowers [18,37]. Through RT-qPCR analysis of *AhTPPs* in different tissues or organs of peanuts, we also observed that almost all genes are highly expressed in both flowers and roots (Figure 6A), which is consistent with the transcriptome data, as well as the results found in wheat [19]. Gene transcription is initiated when transcription factors bind to specific *cis*-elements present in the gene promoter region. Analyzing *cis*-elements in the DNA sequences upstream of the target genes allowed us to predict their transcriptional responses to different stimulations. The discoveries showed that four types of *cis*-elements were recognized, including abiotic stress, phytohormones, and growth-, development- and light response-related elements (Figure 4 and Appendix A). Similar types of abiotic and phytohormone-related *cis*-regulatory elements have been identified in previous studies [34]. RNA-seq and RT-qPCR results indicated that most *AhTPPs* respond to cold and/or drought stress. which was consistent with enriched several low-temperature and drought stress-related (Figure 5 and Figure 6) regulatory *cis*-elements in their promoter regions. Previous studies have found that some *OsTPP* genes are significantly induced under cold or drought stress [21,22]. In cotton, the function of *GhTPP22* under drought stress conditions has also been preliminarily verified [34]. Additionally, four *AhTPPs* (*AhTPP2A*, *AhTPP3B*, *AhTPP4A*, *AhTPP7A*) contain MBSI, flavonoid biosynthetic regulatory elements, suggesting they are involved in flavonoid biosynthesis and plant tolerance to adverse environmental conditions. These studies suggest that *AhTPPs* may play an important role in plant stress resistance. 

Trehalose metabolism has long been thought to be associated with plant stress tolerance [3,38]. Several resurrection plants accumulate high levels of trehalose to withstand desiccation for prolonged periods [39]. TPPs are generally up-regulated under abiotic stress, such as drought and salt stress. In our study, most *AhTPPs* were significantly induced by multiple environmental stresses (Figure 6B,C). Overexpression of peanut *AhTPP1A* and/or *AhTPP5A* enhanced yeast’s tolerance to osmotic stress (Figure 7B). Similarly, overexpression of *AtTPPF* enhanced drought tolerance and trehalose accumulation in Arabidopsis, while disruption of *AtTPPF* led to more H_2_O_2_ under drought conditions [40]. Also, modulating T6P or trehalose contents via targeting *TPPs* has been proven to be an effective approach to increase yields in rice and maize [23,41]. These results indicated that genetic manipulation of *TPP* gene expression regulates T6P, trehalose, and sugar levels, which, in turn, can improve abiotic stress tolerance and enhance crop yield. As our understanding of the specific biological functions of trehalose deepens, it is anticipated to assume a more pivotal role in breeding crops with enhanced stress resistance.

## 4. Materials and Methods

### 4.1. Plant Material and Stress Conditions

The “Zhonghua 215” peanut cultivar was used in stress treatments in this study. “Zhonghua 215” seeds were obtained from the Oil Crops Research Institute (OCRI) of the Chinese Academy of Agricultural Sciences (CAAS). Samples were collected at different stages of groundnut seed growth and development for the analysis of transcriptional expression levels of *AhTPPs* in different tissues or organs. The stress treatments were conducted following the protocols in the previous study [42]. Upon collection, all samples were rapidly frozen in liquid nitrogen and subsequently stored at −80 °C until they were ready for further analysis.

### 4.2. Identification and Characterization of TPP Genes in Peanuts

The cultivated peanut reference genome sequence (cv. Tifrunner) and its annotation were downloaded from the Peanutbase (https://www.peanutbase.org/, accessed on 10 August 2023). Two approaches were used to search for the TPP family members in the peanut genome. Firstly, 23 Arabidopsis and rice TPP protein sequences were used as queries to search in the Peanut Protein Database (https://www.peanutbase.org/, accessed on 10 August 2023) via blastp program (E-value < 1 × 10^−^^10^). Secondly, a Hidden Markov Model (HMM) file of the TPP domain (Trehalose_PPase, PF02358), retrieved from the Pfam database (http://pfam.xfam.org/, accessed on 10 August 2023), was used to search the Peanut Protein Database using HMMER software (v3.1) with an E-value less than 1 × 10^−5^ (http://www.hmmer.org/, accessed on 10 August 2023). After removing redundant sequences, the remaining candidate TPPs were submitted to the CDD (http://www.ncbi.nlm.nih.gov/cdd/, accessed on 10 August 2023) and SMART (http://smart.embl.de/, accessed on 10 August 2023) databases to validate the presence of the TPP domain. Peanut TPP proteins without the domain and those with amino acid lengths under 200 were deleted. Utilizing the ExPASy tool (https://web.expasy.org/protparam/, accessed on 10 August 2023), the molecular weight (Mw), isoelectric points (pI), and grand average of hydropathy (GRAVY) values for the TPPs were computed. Using the PSORT (https://www.genscript.com/psort.html, accessed on 10 August 2023) and CELLO program (http://cello.life.nctu.edu.tw/, accessed on 10 August 2023), predictions were made for the subcellular localization of the TPPs. 

### 4.3. Phylogenetic Analysis

To study the phylogenetic analysis of peanut TPP family members, we retrieved TPP proteins from *A. thaliana* (https://www.arabidopsis.org, accessed on 10 August 2023), *G. hirsutum* (https://cottonfgd.org, accessed on 10 August 2023), *P. tomentosa* (https://phytozome.jgi.doe.gov/pz/portal.html/, accessed on 10 August 2023), *G. max* [33], *A. duranensis* and *A. ipaensis* (https://www.peanutbase.org/, accessed on 10 August 2023). Multiple alignment of TPP protein sequences from both *A. hypogaea* and the other six species was performed using MAFFT v7.490 with default settings [43]. The alignment result was used to construct an NJ phylogenetic tree using the IQ-tree program (v2.2.0) with a bootstrap of 1000 [44]. The tree was visualized using FigTree software (v1.4.4) (http://tree.bio.ed.ac.uk/, accessed on 10 August 2023).

### 4.4. Gene Structure and Chromosomal Distribution

The gff3 annotation files that included the gene structure information were downloaded from the Peanutbase (https://www.peanutbase.org/, accessed on 10 August 2023). To analyze and visualize the exons and introns, we utilized the Gene Structure Display Server (http://gsds.gao-lab.org/, accessed on 10 August 2023) [45]. The conserved motifs of AhTPPs were predicted using the online tool MEME (http://meme-suite.org/tools/meme/, accessed on 10 August 2023) [46] with the following parameters: maximum number of motifs (10), any number of repetitions for site distribution, and optimum motif width (6–50 residues).

### 4.5. Gene Duplication and Collinearity Analysis

To visualize the chromosomal distribution map of *AhTPPs*, we employed MapChart Software v2.32 [47]. To identify *AhTPP* gene duplication events, we utilized the multiple collinear scanning toolkit (MCScanX) with an E-value set to 10^−5^ [48]. The collinearity relationships of *TPPs* between *A. hypogaea* and other species were analyzed and visualized by TBtools software (v1.129) [49]. The ratios of synonymous (Ks) and non-synonymous substitutions (Ka) were calculated by simple Ka/Ks calculator in TBtools. We estimated the divergence time of duplicated gene pairs using the formula: T = Ks/2λ (λ = 8.12 × 10^−9^) [50].

### 4.6. Analysis of Cis-Acting Elements and Expression Profiles of AhTPPs

The 1500 bp upstream sequences of *AhTPP* gene transcription start site were retrieved from peanut genome (https://www.peanutbase.org/, accessed on 10 August 2023), and used for predicting *cis*-elements by PlantCARE (http://bioinformatics.psb.ugent.be/webtools/plantcare/, accessed on 10 August 2023) [51]. Publicly available RNA-seq data were downloaded from the Peanutbase (https://www.peanutbase.org/expression/, accessed on 10 August 2023) to investigate the tissue and developmental expression profiles of *AhTPPs* [52]. The transcriptional responses of *AhTPPs* to abiotic stress were obtained from the Peanut Genome Resource (http://peanutgr.fafu.edu.cn, accessed on 10 August 2023) [29]. Furthermore, our previous transcriptome data (https://www.ncbi.nlm.nih.gov/, access number: PRJNA751249, accessed on 10 August 2023) were used to analyze the expression pattern of *AhTPPs* in peanut seedlings subjected to cold treatment [30]. Transcript levels were estimated using FPKM values (fragments per kilobase of transcript per million fragments mapped), which were converted to log_2_(FPKM+1) to create a hierarchical clustering heatmap and visualized using the R software (https://posit.co/download/rstudio-desktop/, accessed on 10 August 2023) package “pheatmap”.

### 4.7. RNA Extraction and RT-qPCR Based Gene Expression Analysis

Total RNA was extracted using an RNAprep Pure plant RNA extraction kit (TIANGEN, Co., Ltd., Beijing, China) and cDNA was prepared with the M-MLV reverse transcriptase (Thermofisher Scientific, Waltham, MA, USA). RT-qPCR assays were performed according to our previous study [42]. The relative expression levels of *AhTPP* genes across different samples were quantified using the 2^−ΔΔCT^ method [53]. The primer information for RT-qPCR analysis was listed in Appendix A.

### 4.8. Subcellular Localization

The coding sequences of *AhTPP1* and *AhTPP5* were PCR amplified with cDNA as the template, and cloned into the pCambia1300-35S::GFP binary vector. Appendix A provides detailed information on gene access numbers and primer sequences used for cloning purposes. Upon confirming the generation of pCambia1300-35S:*AhTPPs*-GFP vectors through sequencing, they were subsequently introduced into tobacco protoplasts that expressed a red fluorescent nuclear marker (mcherry). The transformation process was carried out using the PEG-mediated method [54]. Following a 48h incubation period, the signals of green fluorescent protein (GFP), red fluorescent protein (RFP), and bright field were visualized using a Zeiss (Carl Zeiss AG, Oberkochen, Baden-Wurttemberg, Geramny) LSM710NLO confocal laser scanning microscope.

### 4.9. Heterologous Expression of AhTPP in Yeast

The coding sequences of *AhTPP1A* and *AhTPP5A* were amplified with the specific primers listed in Appendix A, and then cloned into a pYES2 yeast expression vector using a ClonExpress^®^ MultiS One Step Cloning Kit (Vazyme Biotech Co., Ltd., Nanjing, China). The recombinant plasmid pYES2-AhTPP1A/AhTPP5A and pYES2 empty vector were transformed into a yeast competent cell (strain INVSc1) individually using PEG/LiAc-mediated transformation method. In brief, 10 µL Carrier DNA was denatured at 95 °C for 5 min, then mixed with 2 µg plasmid, 500 µL PEG/LiAc solution, and 100 µL yeast competent cell at 30 °C for 30 min. The mixtures were heated at 42 °C for 5 min, centrifuged, and resuspended with ddH_2_O, then the yeast cells were speeded on synthetic complete SD-Ura plate and incubated at 30 °C for 72 h, finally the positive clones were verified by PCR. For abiotic stress tolerance assays, one positive single clone was inoculated in 10 mL SC-Ura liquid culture medium containing 2% glucose and grown at 30 °C with shaking until the OD_600_ (optical density at 600 nm) reached 0.8. After removing the supernatant, the cells were cultivated in SC-Ura liquid medium containing 2% galactose until OD_600_ = 1.0. Afterwards, the yeast cultures were diluted with SG-Ura medium to four different levels (10^0^, 10^−^^1^, 10^−^^2^, 10^−^^3^), then 2 µL of each diluted culture was spotted onto SG-Ura plates containing varying concentrations of NaCl for salt stress, or mannitol for osmotic stress [31]. Following incubation at 30 °C for three days, photographs of all plates were taken.

## 5. Conclusions

A total of 16 *AhTPP* genes were identified in the cultivated peanut genome and classified into three phylogenetic subgroups. Segmental duplication played a significant role in the expansion of the *AhTPP* gene family. Expression profiles revealed that *AhTPPs* are actively involved in various aspects of peanut growth and development. Furthermore, they also exhibited responsiveness to multiple abiotic stress conditions. AhTPP1A and AhTPP5A were located in both the cytoplasm and nucleus. The heterologous expression of either *AhTPP1A* or *AhTPP5A* enhanced osmotic and salt tolerance in transgenic yeast. This study will not only facilitate further studies on the functions of the *AhTPP* gene family in peanut, but also lay the foundations for elite cultivars with improved tolerance to abiotic stresses.

## Figures and Tables

**Figure 1 plants-13-01056-f001:**
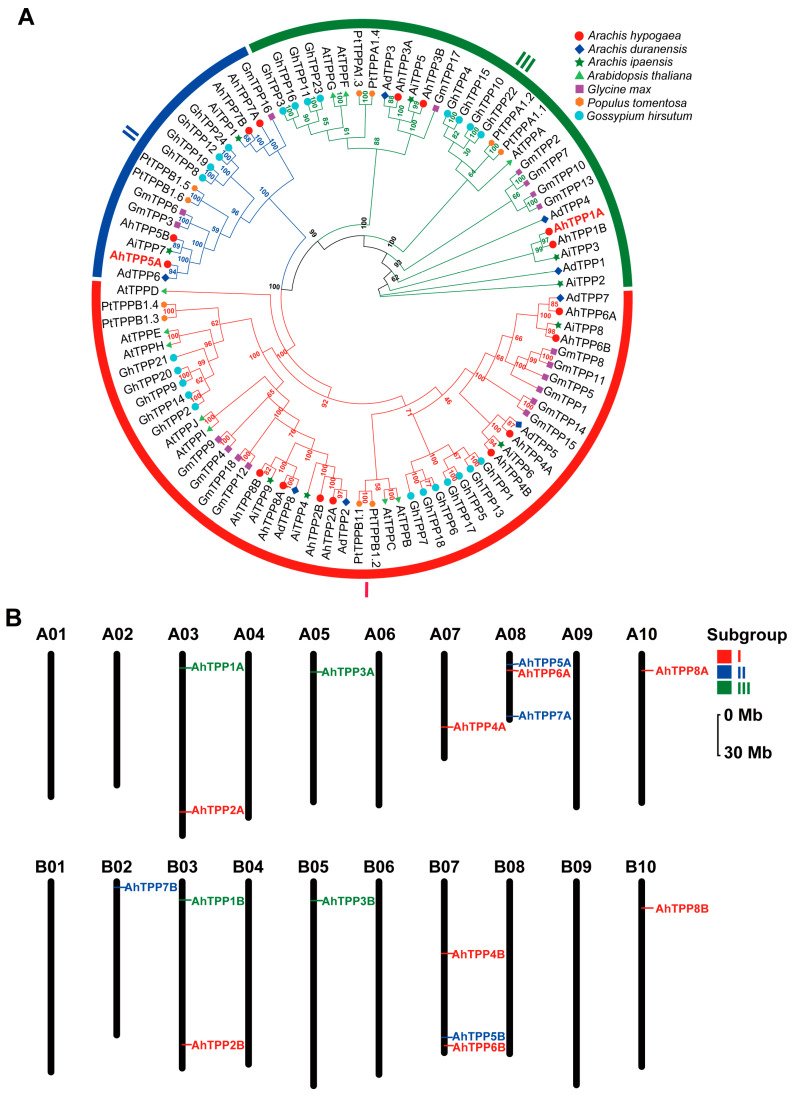
Identification of *AhTPP* gene family members in *Arachis hypogaea*. (**A**) Phylogenetic tree of AhTPPs in *A. hypogaea* and six other plant species. A neighbor-joining phylogenetic tree was generated using 95 TPP protein sequences from *Arachis hypogaea* (Ah), *Arachis duranensis* (Ad), *Arachis ipaensis* (Ai), *Arabidopsis thaliana* (At), *Glycine max* (Gm), *Populus tomentosa* (Pt), and *Gossypium hirsutum* (Gh). TPP members are clustered into three subgroups, distinguished by different colors in the phylogenetic tree. The nodes in the tree display bootstrap values, indicating the confidence level for each cluster. (**B**) Chromosomal distribution of *AhTPPs* in *A. hypogaea*. I–III represent three subgroup. The 30 Mb chromosomal distance is depicted on the vertical axis on the left side of the plot. *AhTPP* genes belonging to distinct phylogenetic subgroups are visually highlighted using different colors.

**Figure 2 plants-13-01056-f002:**
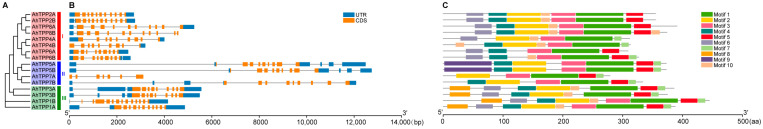
Gene structure and conserved protein motifs of peanut *AhTPP* genes. (**A**) Phylogenetic tree of AhTPPs. IQ-tree software (v2.2.0) was utilized for the construction of the phylogenetic tree, with bootstrap values of 1000 replicates. (**B**) Exon–intron structure of *AhTPP* genes. Coding sequences are represented by orange boxes, untranslated regions by blue boxes, and introns are indicated by black lines. (**C**) Conserved protein motifs in AhTPPs.

**Figure 3 plants-13-01056-f003:**
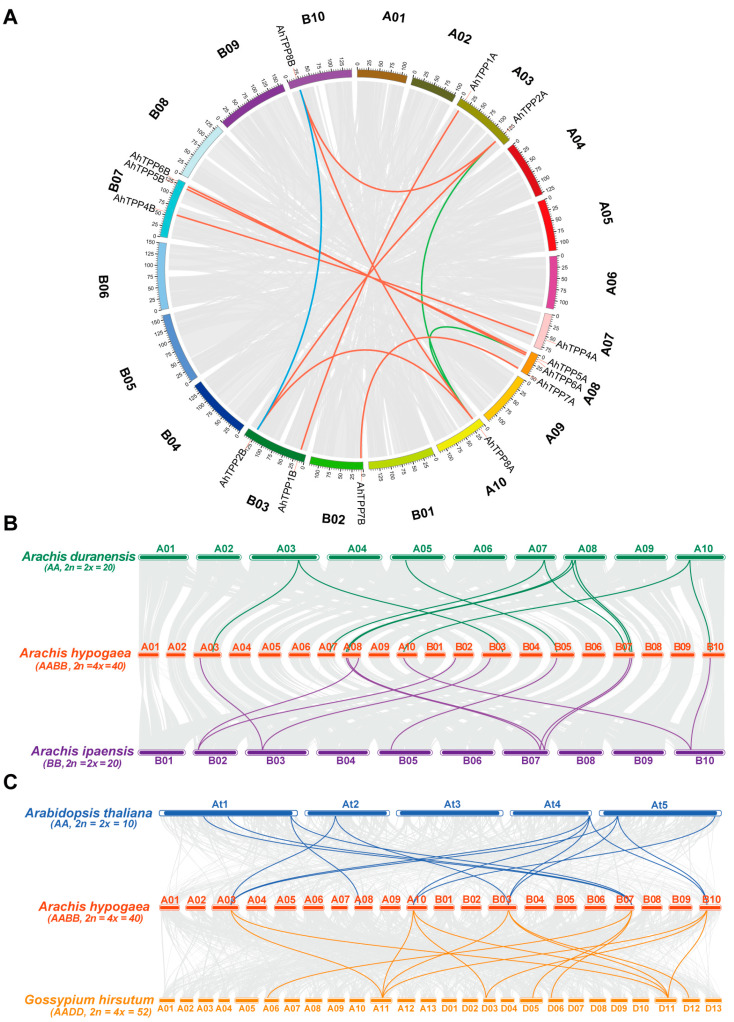
Collinear analysis of *AhTPPs*. (**A**) Collinear analysis of the *AhTPP* gene family in *A. hypogaea*. Chromosomes are represented in the outer circle and are distinguished by different colors. The segmental duplication pairs of *AhTPPs* from different subgenomes (**A**,**B**) are linked by red lines, while those from the same subgenome are linked by green or blue lines. (**B**) Collinear analysis of *TPPs* between *A. hypogaea* and its two diploid ancestors (*A. duranensis* and *A. ipaensis*). (**C**) Collinear analysis of *AhTPPs* with *A. thaliana* and *G. hirsutum*. Gray lines in the background indicate all collinear blocks, while color lines highlight orthologous *TPP* gene pairs.

**Figure 4 plants-13-01056-f004:**
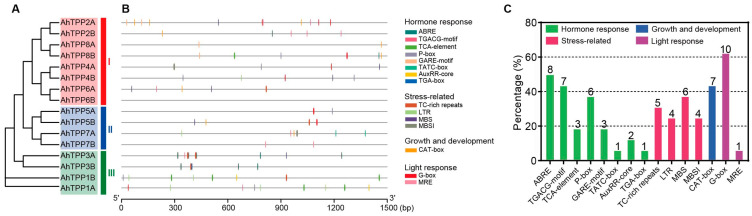
Predicted *cis*-acting elements in the promoter regions of AhTPP genes. (**A**) Phylogenetic tree of AhTPPs. (**B**) Prediction of *cis*-acting elements in the 1500 bp upstream region of the AhTPP gene transcription start site. (**C**) Statistic of *cis*-acting elements in the promoter regions. The values on the top of each bar indicate the counts of AhTPPs containing the corresponding *cis*-acting elements in their promoter regions.

**Figure 5 plants-13-01056-f005:**
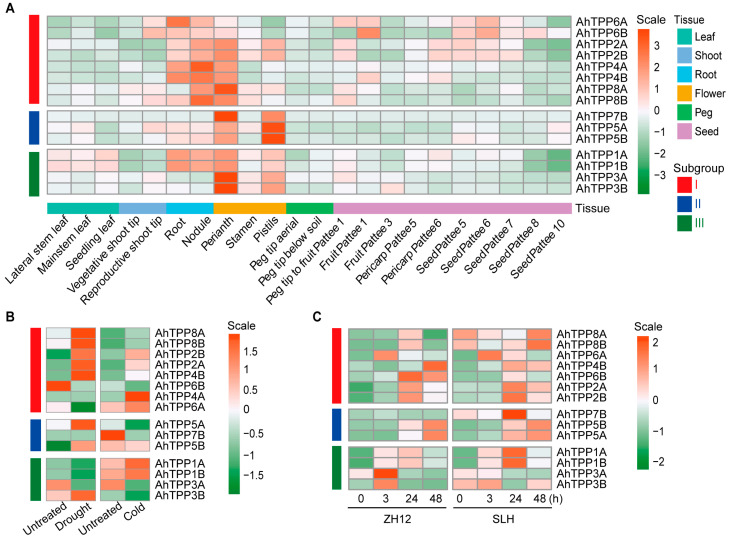
Transcriptional expression profiles of *AhTPPs*. (**A**) Expression patterns of *AhTPPs* in various peanut tissues and at different developmental stages. (**B**) Transcript levels of *AhTPPs* in peanut seedlings exposed to drought and/or cold stress. Untreated indicates the control condition. (**C**) Expression profiles of *AhTPPs* induced by cold treatment at four different time points (0, 3, 24, 48 h) in the two peanut cultivars (ZH12 and SLH). *AhTPPs* from the three phylogenetic subgroups are indicated by different color lines. FPKM values were obtained from the transcriptome results in a publicly available database. *AhTPPs* expression levels were normalized using log2 (FPKM+1), where the heatmap color gradient ranging from orange to green represents high to low gene expression levels.

**Figure 6 plants-13-01056-f006:**
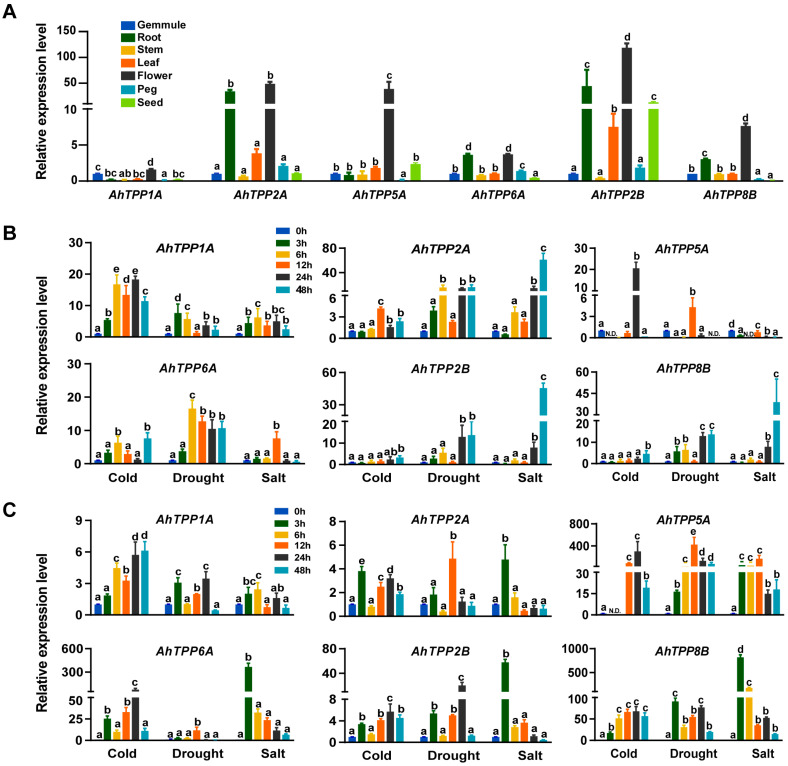
RT-qPCR analysis of *AhTPPs* transcript levels. (**A**) Expression levels of *AhTPPs* in different peanut tissues. (**B**) Expression levels of *AhTPPs* in peanut leaves under different stress conditions. (**C**) Expression levels of *AhTPPs* in peanut roots under different stress conditions. Error bars indicate the standard error of three biological replicates. In a bar chart, the different letters represent significant differences between the samples. N.D. stands for “not detected” in the sample, indicating that the target gene was not detected.

**Figure 7 plants-13-01056-f007:**
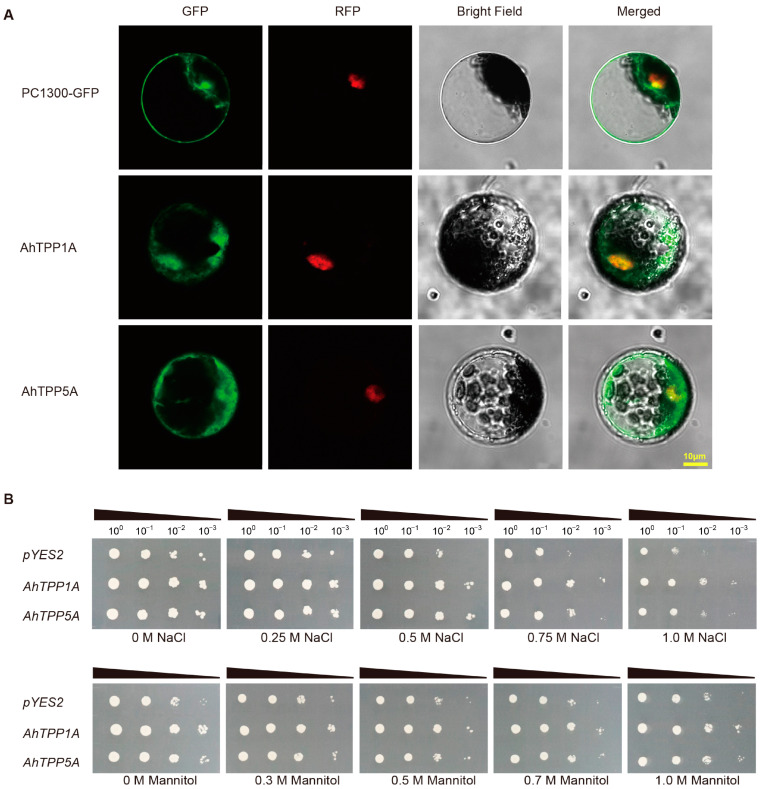
Subcellular localization and functional analysis of AhTPPs. (**A**) Subcellular localization analysis of AhTPPs. PC1300-35S::AhTPP1A-GFP, PC1300-35S::AhTPP5A-GFP, and PC1300-35S::GFP (empty vector control) were transformed into tobacco mesophyll protoplasts expressing a red fluorescent nucleus marker (RFP). (**B**) Overexpression of *AhTPP* enhances tolerance to drought and salt stress in transgenic yeast. Serial dilutions of yeast cells harboring *AhTPP* or pYES2 (empty vector control) were spotted onto SG-U solid medium containing different concentrations of NaCl (salt) or mannitol (osmotic) for stress tolerance assays.

**Table 1 plants-13-01056-t001:** Data of the 16 *TPP* genes identified in *Arachis hypogaea*.

Gene Name	Gene ID	Chr	Protein Length (aa)	pI	Molecular Weight (kDa)	GRAVY	Subcellular Location
*AhTPP1A*	Arahy.FIK9JS.1	A03	388	6.57	43.21	−0.275	cytoplasm, nuclear
*AhTPP2A*	Arahy.P5P8C7.1	A03	355	9.08	39.93	−0.392	apoplasm
*AhTPP3A*	Arahy.51Y4LP.1	A05	386	6.22	43.30	−0.381	cytoplasm
*AhTPP4A*	Arahy.J3QXZL.1	A07	312	9.31	35.41	−0.366	cytoplasm
*AhTPP5A*	Arahy.0FY2NM.1	A08	373	9.22	42.41	−0.332	cytoplasm, nuclear
*AhTPP6A*	Arahy.B1753N.1	A08	309	9.07	34.90	−0.348	apoplasm
*AhTPP7A*	Arahy.B2J19D.1	A08	279	7.79	31.69	−0.295	cytoplasm
*AhTPP8A*	Arahy.SS1VDK.1	A10	391	9.04	43.92	−0.353	apoplasm
*AhTPP1B*	Arahy.CK7SVG.1	B03	445	7.59	49.87	−0.256	apoplasm
*AhTPP2B*	Arahy.FJ2B1F.1	B03	355	9.22	39.95	−0.393	apoplasm
*AhTPP3B*	Arahy.5Q8BGE.1	B05	375	6.03	42.05	−0.410	cytoplasm
*AhTPP4B*	Arahy.DP0G5T.1	B07	313	9.48	35.21	−0.296	apoplasm
*AhTPP5B*	Arahy.G2WE73.1	B07	373	9.33	42.31	−0.343	cytoplasm
*AhTPP6B*	Arahy.KW1U5A.1	B07	327	9.18	36.74	−0.378	apoplasm
*AhTPP7B*	Arahy.U769C2.1	B02	334	5.49	38.13	−0.375	cytoplasm
*AhTPP8B*	Arahy.S15BQI.1	B10	374	9.12	41.95	−0.472	apoplasm

Chr, chromosome; aa, amino acids; pI, isoelectric points; GRAVY, grand average of hydropathy.

## Data Availability

Data presented in this paper are contained within the article and Appendix A.

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
