# Peer review of "Genome-Wide Analysis of Trehalose-6-Phosphate Phosphatase Gene Family and Their Expression Profiles in Response to Abiotic Stress in Groundnut"

_plants, 2024, doi:10.3390/plants13081056_

Round 1

Reviewer 1 Report

Comments and Suggestions for Authors

The manuscript presents a comprehensive study on the identification, characterization, and expression analysis of the TPP gene family in groundnut. Even though the manuscript covers important aspects related to gene family analysis, gene expression profiling, and functional characterization, there are several areas that could be improved and some drawbacks that need to be addressed before considering it for publication.

The manuscript lacks a clear structure, making it challenging for readers to follow the flow of information. It would benefit from a more structured organization to guide the reader through the different components of the study.

The abstract provides a concise summary of the study but could be improved by including more specific details about the methodology, results, and impacts of the findings. It would be beneficial to mention the key findings and contributions of the study more explicitly in the abstract to attract the interest of potential readers.

The introduction provides a comprehensive background on trehalose metabolism and the importance of TPP genes but could be more concise. It would benefit from a clearer statement of the research objectives and hypotheses.

Condense the methodology sections by removing redundant information and focusing on essential details.

The results section provides a detailed description of the identification, characterization, and expression analysis of the AhTPP gene family. However, it lacks clarity and conciseness, making it challenging to extract the key findings.

The discussion section provides a thorough interpretation of the results and their implications. However, it could be more focused and structured to highlight the key findings and their significance. It would be beneficial to compare the findings of this study with previous research in the field and discuss any novel contributions to the existing knowledge.

The references cited in the manuscript are appropriate and relevant to the study. However, there are inconsistencies in citation formatting and numbering that should be corrected for accuracy and consistency.

In conclusion, although the manuscript presents valuable understandings about the genome-wide analysis of the AhTPP gene family in groundnut, there are several areas that need improvement before it can be considered suitable for publication. Addressing the identified drawbacks and suggestions for improvement would enhance the clarity, quality, and impact of the manuscript.

Comments on the Quality of English Language

Though the English language used in the manuscript is generally understandable, there are instances of awkward phrasing, grammatical errors, and repetition that could be improved for clarity and readability. Some sentences are overly complex and could be simplified without losing essential information. There are inconsistencies in terminology usage and formatting that should be addressed for clarity and consistency throughout the manuscript.

Reviewer 2 Report

Comments and Suggestions for Authors

In this manuscript, Liu and collaborators perform a genome-wide analysis of the trehalose-6-phosphate phosphatase (TTP) gene family in cultivated groundnut (Arachis hypogaea). They identified a total of 16 putative AhTTP genes which were classified into three major phylogenetics subgroups, and determined their chromosomal locations, gene structure, collinearity in A. hypogea, as well as between A. hypogea and other plant species. Furthermore, the in silico and RT-qPCR analyses of AhTTP gene expression revealed that AhTTP transcript levels varied in different tissues, developmental stages and under multiple abiotic stress conditions. Finally, the authors found that heterologous overexpression of AhTTP1A and AhTTP5A, both located in the cytoplasm and nucleus, enhanced stress tolerance in yeast.

This work has the novelty that, for the first time, the study of the TPP gene family in groundnut is addressed. Besides, the results of this paper may allow advancing in the knowledge of the participation of AhTTP genes in development, growth and tolerance to abiotic stress in groundnut. The bioinformatics and experimental approaches followed are suitable for studying a gene family for the first time in a plant species. However, in my opinion, more experimental work is lacking. For instance, the authors could have studied by RT-qPCR in different groundnut tissues/organs, the expression of the genes shown in Figure 6, as they did in response to abiotic stress.

I also have some additional comments and suggestions that I discuss below:

-        The term heterogeneous is used instead of heterologous (see pages 2 and 12).

-        The size and resolution of some figures should be increased. Especially figures 1, 2 and 6.

-        Page 4, line 154. “discovered” should be replaced by “identified” or “found”.

-        I suggest including the names of groundnut genes displaying collinearity with Arabidopsis and G. hirsutum genes at the end of page 5.

-        Figure 2A. Bootstrap values should be mentioned in the legend.

-        Section 2.6: It should be noted that almost all TTP genes show high expression in perianth, irrespectively of the subgroup.

-        Section 2.6: The criteria followed to select the AhTTP genes studied by RT-qPCR should be explained.

-        The millions of years values shown in Table S2 are not mentioned in the article.

-        Throughout the manuscript, qRT-PCR should be replaced with RT-qPCR.

-        Section 2.7. It can be read: Similarly, the transgenic yeast harboring AhTPP1A or AhTPP5A showed more resistance to mannitol-mediated drought stress than empty vector, especially when the cultures were diluted up to 10-3. I do not agree with this statement, since the yeast transformed with the empty control vector (pYES2) already shows in the control (0 mM mannitol) much lower growth than the yeast transformed with any of the AhTPP genes. Therefore, the mannitol results should be taken with caution.

-        Discussion. It can be read: Bioinformatic predication from online tools suggested that most of AhTPPs were located in cytosol, which appeared overlapped with the localization of T6P. First, replace “predication” with “prediction”. Second, I wonder if the AhTTP1A and 5A proteins contain a NL signal. Moreover, GFP signal from the 35S::GFP control transgene was apparently also detected in the nucleus, according to the merged picture of Figure 7A. Therefore, I think that it cannot be ruled out that the nuclear localisation of AhTPP1A and 5B could be, at least in part, an artefact produced by overexpression of the gene by a 35S promoter.

-        Discussion. It can be read:  all the peanut AhTPPs were observed to be expressed in roots. However, according to Figure 5, AhTPP8A, 7B, 3A and 3B are not expressed whatsoever in roots.

-        Conclusions. Replace “and responded to multiple abiotic stress responses” with “and responded to multiple abiotic stress conditions”.

Comments on the Quality of English Language

In my opinion the English of the article is adequate and allows the work to be read and understood well. However, there are some mistakes that I point out in my report that should be corrected prior acceptance. 

Reviewer 3 Report

Comments and Suggestions for Authors

Reviewing comments on the article entitled « Genome-wide analysis of trehalose-6-phosphate phosphatase gene family and their expression profiles in response to abiotic stress in ground nut »

The authors identified in silico the genes of the TPP family in groundnut. After a genetic analysis, they looked at the expression profiles under different abiotic stresses in database. A selection of TPP members were used for qRT-PCR analysis and two of them were used for subcellular localization and functional analysis in yeast.

As a general comment, the experiments are well presented, the English writing is of good quality and the amount of work is worth publishing. However, the novelty is poor since all the information were already known is other plant species. The discussion is just a list of results that are in accordance with what was already published.

Below is the list of the different points that should be addressed prior publication:

-       In L168, a divergence time is mentioned without any reference to any data. It seems to come out of nowhere. This point should be explained and discussed in the discussion section as well as the meaning of a purifying evolution for the TPP family.

-       Fig. 5B: the control condition (untreated) is indicated as “CK”. This is confusing since CK is generally used for the hormone cytokinin. Therefore, the legend of CK(drought) or CK(cold) should be renamed as “untreated”.

-       There’s no explanation for the choice of the 6 AhTPP selected in the qRT-PCR experiment (L235). Why these TPPs and no others?

-       Why the authors claims “as expected” for cytoplasmic and nuclear localization of AhTPP1A and 5A since the prediction was cytoplasmic only (L261)? This point should be discussed. Furthermore, since some AhTPPs have a predicted location in apoplasm, the role of TPPs in this extracellular compartment should be discussed.

-       There’s no statistical analysis in this qRT-PCR (fig. 6). The significant difference between each time point should be presented.

-       In fig. 7, the term “drought stress” used for treatment applied on yeast is abusive. Only an osmotic stress is applied. Drought stress can be applied on plants but not on yeasts. This should be corrected. As a consequence, it is not possible to directly compare the stress tolerance in yeast to the one in plant. The osmotic stress tolerance in yeast is just a suggestion of a possible same effect in plant since this application of osmotic agent such as mannitol (or NaCl) just mimics the drought stress. Thus, the text in the discussion (L339-340) should be moderated accordingly.

-       The discussion section is to short and should be developed. For example, the specificity or special interest for peanut species in comparison to other species should be presented.

-       In the Mat&Met section, the description of yeast experiment (paragraph 4.9)  is too light. Information on the yeast transformation method has to be explained as well as the OD600 used for the first dilution.

Minor points:

-       Table 1: molecular weight is in kDa and not in KDa.

A subcellular location in periplasm is inappropriate since plants haven’t periplasm. I presume that apoplasm was meant. This has to be corrected.

AhTPP7B is not in italic in the first column.

-       L112: the word “be” is missing in “…was calculated to be between…”

-       L147: “Among of them” should be corrected as “Among them”

-       L152: idem

-       L200: the word “in” is missing in “…were possibly involved in multiple hormone…”

-       L256: “that” should be added in the sentence “Given that the two…”

-       L447: the yeast strain name should be more clearly indicated “(strain INVSc1)” instead of just “(INVSc)”.

-       L459: “Heterogenous” is misused here. Write “Heterologous” instead.

-       L460: the sentence is not correct and should be rephrased as “This study will not only facilitate further studies on the functions of…”

Comments on the Quality of English Language

The English language is of good quality and required only minor editing revision.

Round 2

Reviewer 1 Report

Comments and Suggestions for Authors

A few minor grammatical errors were still spotted, which can easily be rectified during the proofreading of the final galley, if the manuscript is accepted.

Comments on the Quality of English Language

A few minor grammatical errors were still spotted, which can easily be rectified during the proofreading of the final galley, if the manuscript is accepted.

Author Response

Comments and Suggestions for Authors

A few minor grammatical errors were still spotted, which can easily be rectified during the proofreading of the final galley, if the manuscript is accepted.

--Response: Thanks, we have checked the whole manuscript and modified some grammatical errors. The changes have been tracked.

Comments on the Quality of English Language

A few minor grammatical errors were still spotted, which can easily be rectified during the proofreading of the final galley, if the manuscript is accepted.

--Response: Revised.

Reviewer 2 Report

Comments and Suggestions for Authors

Authors have successfully addressed most of my concerns and suggestions, and the manuscript has been improved. Notwithstanding, I still have a few comments that should be addressed before final acceptance.

1. Lines 283-288. It can be read in the new version of the manuscript: “Although it appeared that transgenic yeast harboring AhTPP1A or AhTPP5A grew a bit better than the yeast with empty vector (pYES2) under control condition (0 M mannitol), the transgenic yeast showed much more resistance to mannitol-mediated osmotic stress than empty vector with the concentration of mannitol increasing to 1.0 M, as no yeast cell with pYES2 was observed on the 1.0 M mannitol plate when the cultures were diluted up to 10-3”. However, I would not say “much more resistance to mannitol-mediated osmotic stress than empty vector” but “slightly more tolerant to mannitol-mediated osmotic stress than empty vector”. I consider that differences shown on Figure 7B do not allow to state that transgenic yeast showed much more resistance to mannitol-mediated osmotic stress than empty vector.

2. I still believe that it cannot fully be ruled out that the nuclear localisation of the AhTPP1A and AhTPP5A proteins may likely be due to overexpression of the AhTPP1A and AhTPP5A genes, since the GFP protein from the 35S:GFP construct is also localised in the nucleus. In line with this, the authors themselves indicate in response to my request, that the AhTPP5A protein lacks any NL. Authors should take this into account.

3. To my request that, according to Figure 5, AhTPP8A, 7B, 3A and 3B are not expressed whatsoever in roots, authors answered that “the heatmap (Figure 5A) shows the relative expression levels of AhTPPs which are normalized using log2 (FPKM+1), with gradient green color representing low gene expression levels. As shown in Figure 5A, the expression levels of AhTPP8A, 7B, 3A and 3B were relatively low in root, but still expressed in root. Their expression levels (FPKM values) among different tissues were listed in Table S7”. I agree with this answer, but authors should at least state that “all the peanut AhTPPs were observed to be expressed in roots but to different extent. Thus, AhTPP8A, 7B, 3A and 3B genes were expressed at very low levels, whereas others, such as AhTPP6A, AhTPP4A, AhTPP4B, AhTPP1A and AhTPP1B were more highly expressed”.

Comments on the Quality of English Language

Minor editing of English language required.

Round 3

Reviewer 2 Report

Comments and Suggestions for Authors

I find the authors' answers convincing. I would just like to add a few comments:

- Lines 27-28: Replace “Subcellular localization analysis showed that AhTPP1A and AhTPP5A were located in both the cytoplasm and the nucleus” with “Subcellular localization analysis showed that AhTPP1A and AhTPP5A were likely located in both the cytoplasm and the nucleus”

-        Line 297: Replace “tolerant” with “tolerance”.

-        Line 360: Replace “localized” with “likely localized”.

-     Line 289: Replace “These findings suggest that the two AhTPPs are located in both cytoplasm and nucleus” with “These findings suggest that the two AhTPPs are likely located in both cytoplasm and nucleus”.
